# Evaluation of T2-Weighted MRI for Visualization and Sparing of Urethra with MR-Guided Radiation Therapy (MRgRT) On-Board MRI

**DOI:** 10.3390/cancers13143564

**Published:** 2021-07-16

**Authors:** Jonathan Pham, Ricky R. Savjani, Yu Gao, Minsong Cao, Peng Hu, Ke Sheng, Daniel A. Low, Michael Steinberg, Amar U. Kishan, Yingli Yang

**Affiliations:** 1Physics and Biology in Medicine IDP, University of California, 650 Charles E Young Drive S, Los Angeles, CA 90095, USA; jonathanpham@mednet.ucla.edu (J.P.); minsongcao@mednet.ucla.edu (M.C.); penghu@mednet.ucla.edu (P.H.); ksheng@mednet.ucla.edu (K.S.); dlow@mednet.ucla.edu (D.A.L.); 2Department of Radiation Oncology, University of California, 200 Medical Plaza Driveway, Los Angeles, CA 90095, USA; rrsavjani@mednet.ucla.edu (R.R.S.); yugao@mednet.ucla.edu (Y.G.); msteinberg@mednet.ucla.edu (M.S.); aukishan@mednet.ucla.edu (A.U.K.); 3Department of Radiology, University of California, 300 Medical Plaza Driveway, Los Angeles, CA 90095, USA

**Keywords:** MR-guided radiation therapy (MRgRT), prostate cancer, urethra, genitourinary (GU) toxicity, treatment planning, contouring

## Abstract

**Simple Summary:**

Stereotactic body radiation therapy (SBRT) has become a standard of care option for prostate cancer patients, utilizing large fractionated dose to shorten treatment times. However, genitourinary (GU) toxicity associated with urethral injury remains prevalent due to non-trivial urethra delineation and sparing at treatment planning and treatment delivery. The aim of our study was to evaluate two optimized urethral MRI sequences (3D HASTE and 3D TSE) with a 0.35T MR-guided radiotherapy (MRgRT) system for urethral visibility and delineation. Among 11 prostate cancer patients, a radiation oncologist qualitatively scored MRgRT 3D HASTE as having the best urethra visibility, superior to CT, clinical MRgRT 3D bSSFP, MRgRT 3D TSE, and similar to diagnostic 3T (2D/3D) T2-weighetd MRI. Moreover, urethra contours from different imaging and clinical workflows demonstrated significant urethra localization variability. Optimized 3D MRgRT HASTE can provide urethral visualization and delineation within an MRgRT workflow for urethral sparing, avoiding cross-modality/system registration errors.

**Abstract:**

Purpose: To evaluate urethral contours from two optimized urethral MRI sequences with an MR-guided radiotherapy system (MRgRT). Methods: Eleven prostate cancer patients were scanned on a MRgRT system using optimized urethral 3D HASTE and 3D TSE. A resident radiation oncologist contoured the prostatic urethra on the patients’ planning CT, diagnostic 3T T2w MRI, and both urethral MRIs. An attending radiation oncologist reviewed/edited the resident’s contours and additionally contoured the prostatic urethra on the clinical planning MRgRT MRI (bSSFP). For each image, the resident radiation oncologist, attending radiation oncologist, and a senior medical physicist qualitatively scored the prostatic urethra visibility. Using MRgRT 3D HASTE-based contouring workflow as baseline, prostatic urethra contours drawn on CT, diagnostic MRI, clinical bSSFP and 3D TSE were evaluated relative to the contour on 3D HASTE using 95th percentile Hausdorff distance (HD95), mean-distance-to-agreement (MDA), and DICE coefficient. Additionally, prostatic urethra contrast-to-noise-ratios (CNR) were calculated for all images. Results: For two out of three observers, the urethra visibility score for 3D HASTE was significantly higher than CT, and clinical bSSFP, but was not significantly different from diagnostic MRI. The mean HD95/MDA/DICE values were 11.35 ± 3.55 mm/5.77 ± 2.69 mm/0.07 ± 0.08 for CT, 7.62 ± 2.75 mm/3.83 ± 1.47 mm/0.12 ± 0.10 for CT + diagnostic MRI, 5.49 ± 2.32 mm/2.18 ± 1.19 mm/0.35 ± 0.19 for 3D TSE, and 6.34 ± 2.89 mm/2.65 ± 1.31 mm/0.21 ± 0.12 for clinical bSSFP. The CNR for 3D HASTE was significantly higher than CT, diagnostic MRI, and clinical bSSFP, but was not significantly different from 3D TSE. Conclusion: The urethra’s visibility scores showed optimized urethral MRgRT 3D HASTE was superior to the other tested methodologies. The prostatic urethra contours demonstrated significant variability from different imaging and workflows. Urethra contouring uncertainty introduced by cross-modality registration and sub-optimal imaging contrast may lead to significant treatment degradation when urethral sparing is implemented to minimize genitourinary toxicity.

## 1. Introduction

Stereotactic body radiation therapy (SBRT) has become a standard of care option for prostate cancer patients, utilizing a large-fractionated dose to shorten treatment times. Recent SBRT reports with large prostate cancer patient cohorts have shown SBRT to have comparable biochemical control and toxicity rates to conventional treatments [1,2,3]. Despite improvements in treatment efficiency, patients are still reporting acute and late gastrointestinal (GI) and genitourinary (GU) toxicities. GU toxicities can arise due to complications along the GU tract [4]. In the past, attention has primarily been focused on sparing the bladder despite the significant contributions of urethral injury to GU toxicities, mainly due to difficulty in localizing the urethra. 

Delineating the prostatic urethra on CT is challenging due to the urethral wall and prostate having the same physical density and average atomic number [5]. Furthermore, on-board cone beam CT (CBCT) in linear accelerators (LINACs) has poor image quality, making it impossible to visualize the urethra during treatment. The prostatic urethra can vary in size, shape, and length from patient to patient, but on average is approximately 3.5 cm long and 0.8 cm wide [6,7]. Currently, there are no consensus guidelines for contouring the urethra. The use of a Foley catheter has been used to localize and visualize the urethra on CT. However, this method must be done prior to each treatment, is invasive and can lead to infection. Additionally, the catheter can also rotate and deform the urethra, resulting in potential organ misalignment during each treatment delivery [8,9,10]. As a result, some physicians choose to contour the prostatic urethra on the planning CT based on prior experience and knowledge. However, this is unreliable and can be inconsistent between radiation oncologists and centers.

Alternatively, MRI provides superior soft-tissue contrast and proper MRI sequences may be used to improve prostatic urethra conspicuity. T2-weighted (T2w) MRIs can make the urethra appear more hyperintense [4], and are currently used in radiation therapy urethra contouring by registering diagnostic T2w MRI to the planning CT. However, the associated MR to CT registration can be challenging due to differences in tissue contrast between the two imaging modalities, as well as the potentially different shape and location of the urethra on MRI and CT, which is often acquired on different days with different patient position [11].

Delineating the urethra enables various urethra sparing techniques for reducing GU toxicities. Urethra sparing is most commonly implemented by limiting hotspots in the urethra as encouraged in PACE-B [1] and ongoing NRG-GU 005 trials. This method lowers GU toxicity and avoids loss of efficacy. Moreover, limiting hotspots can enable dose escalation elsewhere for aggressive disease, and improve biochemical control while maintaining acceptably low rates of toxicity [12]. Urethra dose de-escalation has also been attempted [13,14,15] but is not commonly used due to the high risk of recurrence at the periurethral areas [16].

In a study by VU Medical Center in Amsterdam (VUMC), prostate cancer patients, undergoing urethra-sparing SBRT using MR-guided radiation therapy (MRgRT), showed lower rates of GU toxicity [13,14,15]. Besides tighter planning target volume margins and on-line adaptation enabled by on-board MRI, the urethra was also delineated and used for urethra sparing with dose de-escalation. In their study, the urethra was contoured on one sagittal MR slice from the balanced steady-state free precession (bSSFP) planning MRI and expanded isotopically by 2 mm. However, the T2/T1 weighted contrast of bSSFP is not ideal for urethra visibility and single-slice urethral contouring is limited and can miss the full extent of the urethra.

In this study, we sought to optimize two MRI sequences, 3D half-Fourier acquisition single-shot turbo spin echo (HASTE) and 3D turbo spin echo (TSE) on a commercial MRgRT system for visualization and multi-slice delineation of the prostatic urethra within an MRgRT prostate SBRT workflow. Imaging on an MRgRT system, as opposed to a diagnostic MRI scanner, allows the patient to be in treatment position and avoids additional systematic uncertainties. HASTE and TSE are T2-weighted MR sequences and can directly provide urethral contrast within the prostate, whereas the clinical bSSFP provides a mix signal (T2/T1 contrast) with little urethral contrast. The tradeoff for our proposed sequence is the relatively lower signal-to-noise ratio (SNR), which can degrade urethral visualization, thus urethral contrast and SNR must be optimized by sequence parameters adjustments for adequate urethral localization. We compared prostatic urethra contours in five different workflows: (1) CT-based planning based on CT only (CT-1), (2) CT-based planning based on CT and registered diagnostic T2w 3T MRI (CT-2), (3) MRgRT-based planning with the proposed optimized urethra 3D HASTE (MRgRT-1), (4) MRgRT-based planning with the proposed optimized urethra 3D TSE (MRgRT-2), and (5) MRgRT-based planning with clinical bSSFP MRI (MRgRT-3). 

## 2. Materials and Methods

### 2.1. Imaging and Subject Cohort

Eleven prostate cancer patients undergoing radiation therapy between February 2020 and June 2020 were included Each patient provided written consent prior to the study. Patient planning CT and diagnostic 3T T2w MRI were acquired prior to treatment. For each patient, the proposed 3D HASTE and 3D TSE sequences were used to image the urethra on a 0.35T MRgRT system (MRIdian, ViewRay). Of the 11 patients, eight patients (Patients 1–8) were imaged immediately after one of their treatment fractions and three patients (Patients 9–11) were imaged right after simulation. Additionally, a clinical bSSFP scan was acquired on the MRgRT system, which is currently used for MRgRT treatment planning and patient setup. Although images were acquired at different times, all but the diagnostic MRI were acquired with the patient staying in the treatment position.

Pelvic CTs were acquired on a 16-slice CT scanner (Sensation Open, Siemens Medical Solutions, Erlangen, Germany) using 120 kVp and 400 mA. CT slice thickness was 1.5–3 mm and in-plane resolution was 0.90 × 0.90 mm^2^–1.27 × 1.27 mm^2^. Diagnostic MRIs were acquired either in 2D or 3D. The scan parameters for the diagnostic T2w MRI, optimized MRgRT HASTE, optimized MRgRT TSE, and clinical MRgRT bSSFP are shown in Table 1.

Both 3D HASTE and 3D TSE were qualitatively optimized for urethra visualization on low-field MRgRT using four healthy male volunteers. Echo time (TE), relaxation time (TR), and voxel size were tuned to provide urethra contrast while maintaining sufficient SNR. Additionally, the number of averages was tuned to increase image SNR and minimize motion/ghosting artifacts, while maintaining acceptable scan times. Figure 1 and Figure 2 show the MRgRT 3D HASTE and MRgRT 3D TSE optimization. Optimization steps for MRgRT 3D HASTE and TSE were similar. Volunteer 1 HASTE 1 and Volunteer 2 TSE 1 (baseline) show a noisy urethra with some urethral contrast. By increasing the number of averages to 6 and 4, the scan SNR improved and the final scan time was increased to 8:06 min and 7:14 min, respectively. Increasing TE to 407 ms (Volunteer 2 HASTE 1) resulted in lower SNR, but greater urethral contrast as T2-weighting increased. Conversely, decreasing TE to 135/133 ms (Volunteer 2 HASTE 2/Volunteer 2 TSE 2), resulted in higher SNR, but less urethral contrast as T2-weighting decreased. Increasing TR to 3000 ms (Volunteer 4 TSE 2) slightly improved SNR, but lowered urethral contrast, as prostate and urethral signal intensity were able to fully recover. Ultimately, TE of 246/250 ms and TR of 1800/2000 ms for MRgRT 3D HASTE/TSE was selected to provide adequate SNR and urethral contrast. Lower resolution (Volunteer 2 HASTE 2 and 3) of 2.0 mm isotropic, compared to 1.5 mm isotropic, provided higher SNR; however, 1.5 mm isotropic was selected to capture the prostatic urethra’s fine structure. Lastly, sagittal acquisition was preferred over axial acquisition for easier urethra visualization and delineation. 

### 2.2. Urethra Contours

A resident radiation oncologist, with over two years of experience, contoured the prostatic urethra on each patient’s CT sim, registered pre-treatment T2w diagnostic MRI, MRgRT 3D HASTE, and MRgRT 3D TSE MRIs, respectively. First, in CT-1 workflow, the resident radiation oncologist only had access to the patient’s CT sim and the prostatic urethra contour was made using anatomical guidelines [17]. Second, in CT-2 workflow, the resident radiation oncologist contoured the prostatic urethra using the patient’s CT sim and the diagnostic T2w MRI which was rigidly registered to the CT. Third, in MRgRT-1 workflow, the resident radiation oncologist contoured the prostatic urethra using the patient’s MRgRT 3D HASTE. Fourth, in MRgRT-2 workflow, the resident radiation oncologist contoured the prostatic urethra using the patient’s MRgRT 3D TSE. Afterwards, an attending radiation oncologist, with over eight years of experience, reviewed and, if necessary, manually edited the resident radiation oncologist’s contours. Lastly, in MRgRT-3 workflow, the attending radiation oncologist contoured the prostatic urethra using the patient’s clinical MRgRT 3D bSSFP.

### 2.3. Evaluation Metrics

The resident radiation oncologist, attending radiation oncologist, and a senior medical physicist with over 15 years of experience qualitatively scored the urethra visibility for each image on a 4-point scale: 1 = no conspicuity; 2 = some conspicuity, urethra can be identified, but not very clear; 3 = good conspicuity, urethra can be identified clearly; 4 = excellent conspicuity.

Based on the urethra conspicuity scores, MRgRT 3D HASTE in MRgRT-1 workflow had the highest score (Table 2) from two out of the three observers and was subsequently used as the reference in quantitative contour evaluation. Specifically, CT, diagnostic MRI, MRgRT 3D TSE, and clinical MRgRT 3D bSSFP were rigidly registered to MRgRT 3D HASTE based on the prostate gland in MIM Software (Cleveland, OH, USA). Afterwards, the contours in CT-1 workflow, CT-2 workflow, MRgRT-2 workflow and MRgRT-3 workflow were compared relative to the contours in MRgRT-1 workflow (MRgRT 3D HASTE) using 95th percentile Hausdorff distance (HD95), mean-distance-to-agreement (MDA), and DICE coefficient. The 95th percentile Hausdorff distance was calculated as the 95th percentile surface distance between contours and was chosen as it is more stable to small outliers.

Additionally, prostatic urethra contrast-to-noise ratios (CNR) were calculated for each image as follows:(1)$${\mathrm{CNR}}_{\mathrm{Prostatic}\;\mathrm{Urethra}\;}=\frac{\left|\mathrm{Prostatic}\;\mathrm{urethra}\;\mathrm{mean}\;\mathrm{intensity}\;-\;\mathrm{Surrounding}\;\mathrm{prostate}\;\mathrm{mean}\;\mathrm{intensity}\right|}{\mathrm{Background}\;\mathrm{noise}}.$$

Prostatic urethra mean intensity was evaluated based on the radiation oncologist’s contour. Surrounding prostate mean intensity was evaluated based on a prostate ring contour encompassing the prostatic urethra. The prostate ring contour was made by expanding the prostatic urethra contour 1 cm isotropically, then subtracting the original urethra. Both the prostatic urethra and prostate ring contour were cropped to remain within the clinical prostate contour, which was originally made on the planning image and registered/transferred to each image. The entire clinical prostate contour for prostate mean intensity calculation was not used because the clinical prostate contour often extended into the bladder and would artificially increase the mean value. The background noise was measured as the standard deviation of the prostate ring contour.

Prostatic urethra qualitative and CNR results were compared using Wilcoxon signed-rank test with a significance level of 0.05.

## 3. Results

Table 2 shows the attending radiation oncologist’s (Observer 1–OBS1), resident radiation oncologist’s (Observer 2—OBS2), and senior medical physicist’s (Observer 3—OBS3) qualitative urethra visibility scores for all patient images. All observers scored CT urethra visibility a 1. For Observer 1 and 3, the qualitative scores for MRgRT 3D HASTE were scored significantly higher than CT and clinical MRgRT 3D bSSFP. Observer 1 scored MRgRT 3D HASTE significantly higher than MRgRT 3D TSE, but Observer 3 did not. Furthermore, Observer 2 scored MRgRT 3D HASTE significantly higher only for CT and not for the other MRI techniques. Two-dimensional (2D) and 3D diagnostic T2w MRI’s qualitative scores were not significantly different from MRgRT 3D HASTE for all observers. MRgRT 3D HASTE was scored highest for two out of three observers and was used as reference for quantitative evaluation. 

Figure 3 shows Patient 11’s (Figure 3a) planning CT, (Figure 3b) 2D diagnostic T2w MRI, (Figure 3c) MRgRT 3D HASTE, (Figure 3d) MRgRT 3D TSE, and (Figure 3e) clinical MRgRT 3D bSSFP. In the planning CT, there is no tissue contrast between the urethra and prostate. In the MRI scans, the contrast is improved in general, but urethra visibility varies in different MRI sequences. Notably, diagnostic MRI and MRgRT 3D HASTE and TSE showed less motion/ghosting artifacts, compared to MRgRT 3D bSSFP (blue arrow).

Figure 4 shows Patient 7’s CT and MRgRT 3D HASTE fused image with urethra contour based on CT-1: CT only (yellow), CT-2: CT + diagnostic T2w MRI (green), MRgRT-1: MRgRT 3D HASTE (red), MRgRT-2: MRgRT 3D TSE (blue), and MRgRT-3: clinical MRgRT 3D bSSFP (purple).

Figure 5 shows CT-1, CT-2, MRgRT-2 and MRgRT-3 workflows’ prostatic urethra contours’ HD95, MDA, and DICE coefficient relative to workflow MRgRT-1’s prostatic urethra contour. The mean HD95s for workflow CT-1, CT-2, MRgRT-2, and MRgRT-3 were 11.35 ± 3.55 mm, 7.62 ± 2.75 mm, 5.49 ± 2.32 mm, and 6.34 ± 2.89 mm, respectively. Similarly, the mean MDAs were 5.77 ± 2.69 mm, 3.83 ± 1.47 mm, 2.18 ± 1.19 mm, and 2.65 ± 1.31 mm, and the mean DICE coefficients were 0.07 ± 0.08, 0.12 ± 0.10, 0.35 ± 0.19, and 0.21 ± 0.12. Overall, the prostatic urethra contours showed great variance between the different workflows. Urethra contours from the three MRgRT MRIs acquired in common imaging sessions (MRgRT-1, MRgRT-2, and MRgRT-3) showed the smallest variances.

Figure 6 shows prostatic urethra CNRs. The mean CNRs for CT, diagnostic T2w MRI, MRgRT 3D HASTE, MRgRT 3D TSE, and clinical MRgRT 3D bSSFP were 0.07 ± 0.05, 0.25 ± 0.14, 0.44 ± 0.25, 0.39 ± 0.29, and 0.24 ± 0.14, respectively. The CNR for MRgRT 3D HASTE was significantly greater than CT (*p* < 0.001), diagnostic T2w (*p* < 0.042), and clinical MRgRT 3D bSSFP (*p* < 0.014), but was not significantly different from MRgRT 3D TSE (*p* = 0.465).

## 4. Discussion

This study evaluated five workflows with the intention to delineate the prostatic urethra, utilizing three MRgRT workflows with different MR pulse sequences on a 0.35T MRgRT system and two conventional CT-based clinical workflows. The two new urethral T2w MR pulses sequences, MRgRT 3D HASTE and 3D TSE, acquired 1.5 mm isotropic resolution images in 7 to 8 min. Based on Observer 1 and 3, both urethral MRgRT sequences were markedly better than current clinical MRgRT 3D bSSFP and CT for urethra visualization. Clinical 3D bSSFP has its intrinsic advantages of fast speed and high SNR, which can be used to acquire 3D volumetric MRI within a very short time. Optimized MRgRT 3D HASTE and TSE showed less motion/ghosting artifacts, originating from subcutaneous fat and periprostatic fat, and better urethra/prostate contrast than clinical MRgRT 3D bSSFP. Thus, the urethral sequences’ intended purpose should be to supplement current clinical MRgRT 3D bSSFP for urethral sparing. However, despite the potential improvement in GU toxicity reduction, the current urethral scan times are long, which can introduce unwarranted organ motion, from bladder or rectum filling, to the treatment planning or setup workflow, potentially degrading the treatment efficacy. Future work will be focused on decreasing the urethral scan time by exploring acceleration strategies, further optimizing the protocol, and utilizing new vendor-improved receiver coils. Moreover, a limitation of the study was a lack of GU toxicity reports. Future work will look to incorporate MRgRT on-board urethral imaging into a urethra sparing study to evaluate its effectiveness in GU toxicity reduction.

No observer was able to visualize the urethra on CT as it provided no urethral contrast. Observer 1 and 3 reported little to no visibility on clinical MRgRT bSSFP, whereas, contrastingly, Observer 2 reported more visibility. Despite this, all observers reported similar high visibility scores for diagnostic MRI and MRgRT urethral scans. Overall, the MRgRT 3D HASTE performed best and was most comparable to diagnostic MRI. Although diagnostic MRI was acquired at higher field strength and is expected to have superior image quality, the low-field MRgRT urethral sequences were able to achieve similar to superior prostatic urethra visualization as it was optimized for urethra visualization Specifically, our low-field MRgRT urethral sequences utilized heavier T2-weighting (larger TE), providing greater urethral contrast as seen in Figure 6. Moreover, diagnostic MRI acquisition is in the axial orientation with large slice thickness and are often acquired in 2D at an oblique angle, making the urethra difficult to be seen.

Although MRgRT 3D HASTE and diagnostic MRI had similar prostatic urethra visibility, the prostatic urethra contours differed significantly. Contour differences may be due to the images being acquired on different days and the patient being in different positions. Furthermore, no strict guidelines were followed to control patient bladder and rectum fullness for diagnostic MRI acquisition, and as a result, urethra location and shape may vary between a patient’s diagnostic MRI and MRgRT 3D HASTE.

MRgRT 3D TSE’s prostatic urethra contour had the best agreement with MRgRT 3D HASTE’s with the highest DICE score and smallest HD95 and MDA values. Both urethra MRgRT scans were acquired on the same day and in the same imaging session, however, there are still significant differences between these two contours, which may be due to different image contrast and potential motion during long acquisition. Furthermore, the prostatic urethra is a small structure, making the oncologist’s contour extremely sensitive to any deviation. Compared to MRgRT 3D TSE, MRgRT 3D HASTE showed less grainy prostate glands, and subsequently easier visualization of urethras. This is also indicated by MRgRT 3D HASTE’s higher CNR. Although MRgRT 3D HASTE’s CNR was superior, its standard deviation was high, indicating inconsistent performance. CNR variance may be due to different prostate patients having varying residual amounts of urine in the prostatic urethra. Additionally, surrounding fat and ghosting artifacts decreases the CNR of the prostatic urethra. Furthermore, the prostatic urethra may be compressed due to nearby prostatic hyperplasia in the transitional zone [18]. Regardless, the prostatic urethra’s lining is histologically different from the surrounding prostate and should be distinguishable on MRI [19]. Future work will focus on implementing fat suppression for more consistent contrast and improving scan technique for motion robustness.

The quantitative results in Figure 5 showed considerable disagreement of prostatic urethra position amongst CT and MR based workflows regardless if MRgRT imaging was done prior to or during treatment course. High urethra contouring accuracy and precision are critical for urethra sparing and radiation therapy efficacy as significant treatment degradation could occur if reduced dose regions were not positioned correctly. One limitation of the study is a lack of a urethra ground-truth to reference. As a result, urethra accuracy could not be confidently reported. Currently, there is no gold-standard ground-truth for the urethra. However, our high MRgRT urethra visibility scores lead us to have higher confidence in the urethra contouring. Furthermore, MRgRT workflow avoids additional cross-modality image registration as the urethra can be reliably drawn on the planning MRI, at the time of simulation or before each treatment. With greater confidence, urethra sparing can be further improved with high visibility urethra on-board imaging for patient setup and pre-beam MRgRT on-line adaptive. Future work will be to design and construct anthropomorphic prostate phantom to study urethra contouring accuracy of our MRgRT urethra MRI sequences. Additionally, future work will be focused on detecting inter-observer and inter-fraction urethra localization variability for determining urethral margin.

One weakness of our study is the small patient cohort. Future work will add more patients to improve the power of the study. Another weakness was the qualitative scoring system, which was subjective. Future work should recruit additional radiation oncologists for more confident scoring and analysis.

## 5. Conclusions

Two 0.35T MRgRT T2w pulse sequences were proposed for urethra visualization and prostatic urethra contouring. MRgRT 3D HASTE provided high contrast and spatial resolution for prostatic urethra delineation. MRgRT workflow avoids cross-modality registration errors and holds the potential of accurate urethra delineation and effective urethra sparing during both initial MRgRT treatment planning and on-line adaptive radiation therapy.

## Figures and Tables

**Figure 1 cancers-13-03564-f001:**
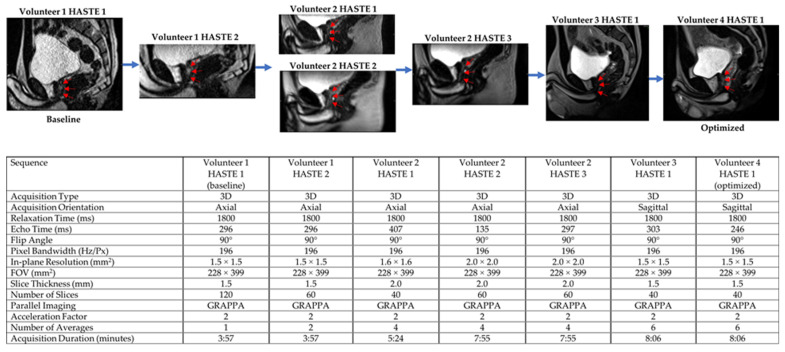
MRgRT 3D HASTE optimization scheme using healthy male volunteers (red arrows pointing towards prostatic urethra).

**Figure 2 cancers-13-03564-f002:**
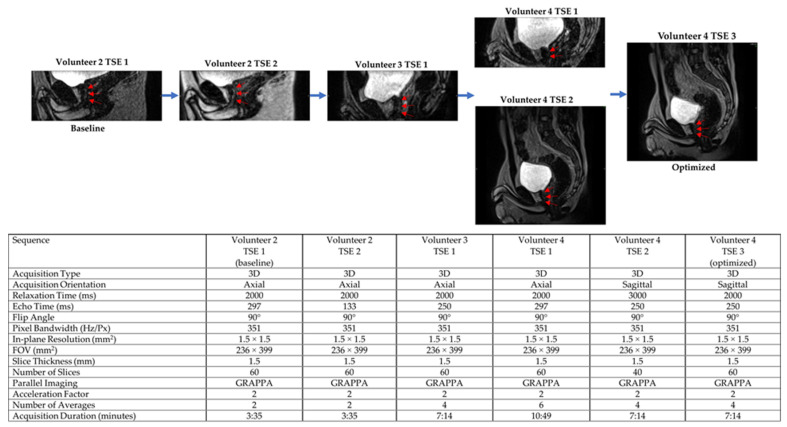
MRgRT 3D TSE optimization scheme using healthy male volunteers (red arrows pointing towards prostatic urethra).

**Figure 3 cancers-13-03564-f003:**
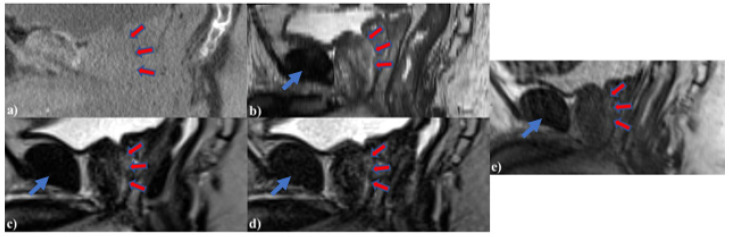
Patient 11’s (**a**) planning CT, (**b**) 2D diagnostic T2w MRI at 3T, (**c**) MRgRT 3D HASTE, (**d**) MRgRT 3D TSE, and (**e**) clinical MRgRT 3D bSSFP (red arrows pointing to the prostatic urethra). Blue arrows showing different amounts of motion/ghosting artifacts in each MRI image. Clinical MRgRT 3D bSSFP showed significant motion/ghosting artifacts.

**Figure 4 cancers-13-03564-f004:**
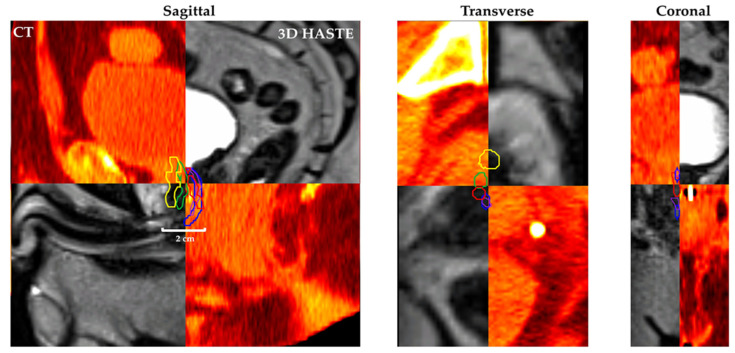
Patient 7’s CT (red/yellow) and MRgRT 3D HASTE (B–W linear) fused image with urethra contour based on workflow (1) CT-1: CT only (yellow), (2) CT-2: CT + diagnostic T2w MRI (green), (3) MRgRT-1: MRgRT 3D HASTE (red), (4) MRgRT-2: MRgRT 3D TSE (blue), and (5) MRgRT-3: clinical MRgRT 3D bSSFP (purple) prostatic urethra contour.

**Figure 5 cancers-13-03564-f005:**
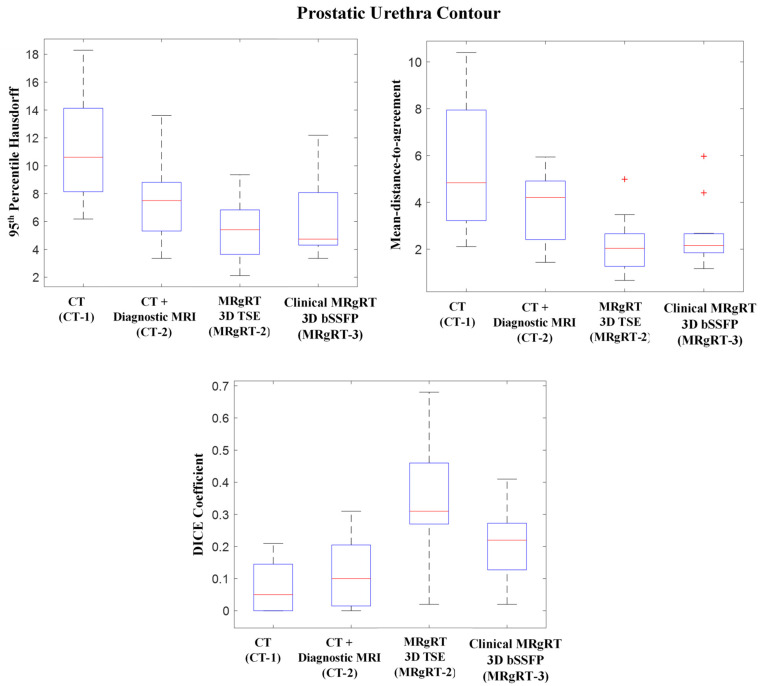
Boxplot of 95th percentile Hausdorff distance, mean-distance-to-agreement, and DICE coefficient for CT (CT-1), CT + diagnostic T2w MRI (CT-2), MRgRT 3D TSE (MRgRT-2), and clinical MRgRT 3D bSSFP (MRgRT-3) prostatic urethra contour relative to MRgRT 3D HASTE’s (MRgRT-1) prostatic urethra contour. CT-based planning showed great variability while MRgRT-based planning showed the most consistency. (Red line = median value, top edge of box = 75th percentile, bottom edge of box = 25th percentile, whiskers = extreme data points (not outliers), red cross = outliers).

**Figure 6 cancers-13-03564-f006:**
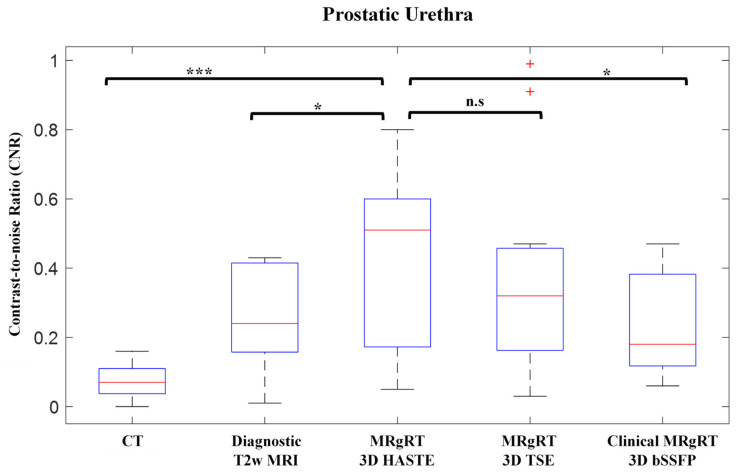
Boxplot of CT, diagnostic T2w MRI, MRgRT 3D HASTE, MRgRT 3D TSE, and clinical MRgRT 3D bSSFP prostatic urethra CNR. Urethral MRgRT scans showed significantly greater prostatic urethra contrast. (Red line = median value, top edge of box = 75th percentile, bottom edge of box = 25th percentile, whiskers = extreme data points (not outliers), red cross = outliers; n.s = not significantly different; ***** = significantly different—* = *p* ≤ 0.05, *** = *p* ≤ 0.001).

**Table 1 cancers-13-03564-t001:** Sequence parameters used for diagnostic T2W (2D and 3D) MRI, MRgRT 3D HASTE, MRgRT 3D TSE, and clinical MRgRT 3D bSSFP.

Sequence	Diagnostic	T2w MRI	MRgRT HASTE	MRgRT TSE	Clinical MRgRT bSSFP
Acquisition Type	2D	3D	3D	3D	3D
Acquisition Orientation	Axial	Axial	Sagittal	Sagittal	Axial
Repitition Time (ms)	3530–6000	2200	1800	2000	3.37
Echo Time (ms)	95–130	205	246	250	1.45
Flip Angle	90–160°	110°	90°	90°	60°
Pixel Bandwidth (Hz/Px)	199–273	315	196	351	535
In-plane Resolution (mm^2^)	0.3 × 0.3–1.3 × 1.3	0.7 × 0.7	1.5 × 1.5	1.5 × 1.5	1.5 × 1.5
FOV (mm^2^)	180 × 180–462 × 399	170 × 170	227 × 400	236 × 399	449 × 499
Phase Encoding Direction	RL	RL	AP	AP	AP
Slice Thickness (mm)	3	1.5	1.5	1.5	1.5
Number of Slices	24–84	60	40	60	192
Parallel Imaging	GRAPPA	GRAPPA	GRAPPA	GRAPPA	None
Acceleration Factor	2–3	2	2	2	N/A
Number of Averages	1–4	2	6	4	1
Acquisition Duration (minutes)	4:00–5:00	7:00	8:06	7:12	1:45

**Table 2 cancers-13-03564-t002:** Observer 1 (OBS1—senior radiation oncologist), observer 2 (OBS2—resident radiation oncologist), and observer 3 (OBS3—senior medical physicist) patient urethra visibility scores for each imaging technique. Qualitative visibility scores: 1 = no conspicuity; 2 = some conspicuity; 3 = good conspicuity; 4 = excellent conspicuity. Wilcoxon signed-rank test statistical significance difference test was used between MRgRT 3D HASTE and different imaging techniques for each respective observer.

Patient	CT			Diagnostic T2w MRI			MRgRT 3D HASTE			MRgRT 3D TSE			Clinical MRgRT 3D bSSFP		
	OBS1	OBS2	OBS3	OBS1	OBS2	OBS3	OBS1	OBS2	OBS3	OBS1	OBS2	OBS3	OBS1	OBS2	OBS3
1	1	1	1	(3D) 2	3	3	3	2	4	3	3	3	2	2	2
2	1	1	1	(3D) 2	2	2	3	2	4	2	3	3	1	3	1
3	1	1	1	(2D) 2	2	2	3	3	3	2	2	3	2	3	1
4	1	1	1	(2D) 2	3	2	3	3	3	3	4	4	2	2	2
5	1	1	1	(3D) 3	3	3	4	3	4	3	4	4	3	3	2
6	1	1	1	(2D) 4	3	4	3	3	4	3	3	3	2	3	2
7	1	1	1	(3D) 4	3	4	3	3	3	2	3	3	2	2	2
8	1	1	1	(3D) 3	3	3	3	3	4	2	3	3	1	2	1
9	1	1	1	(2D) 2	4	3	2	3	3	2	3	3	1	3	1
10	1	1	1	(3D) 4	4	4	3	2	3	2	3	3	1	2	1
11	1	1	1	(2D) 4	4	4	4	4	4	3	4	3	2	3	2
Mean	1.0	1.0	1.0	2.9	3.1	3.1	3.1	2.8	3.5	2.5	3.2	3.2	1.7	2.5	1.5
Standard Deviation	0.0	0.0	0.0	0.9	0.7	0.8	0.5	0.6	0.5	0.5	0.6	0.4	0.6	0.5	0.5
*p*-value	<0.001	<0.001	<0.001	0.732	0.5	0.234	-	-	-	0.02	0.219	0.219	<0.001	0.375	<0.001

## Data Availability

The data in this study are not publicly available.

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
