# Peer review of "Evaluation of T2-Weighted MRI for Visualization and Sparing of Urethra with MR-Guided Radiation Therapy (MRgRT) On-Board MRI"

_cancers, 2021, doi:10.3390/cancers13143564_

Round 1

Reviewer 1 Report

This manuscript addresses the use of T2-weighted MRI for visualization of urethra in MRgRT On-board MRI. The topic is of relevance and of interest for readers involved in MRgRT. The manuscript is worth of publication although I think there are some points which might be addressed:

  1. My main major concern is related to the conspicuity score, as is something relative and it has not been properly described which criteria was followed. Furthermore, the results thereafter are based on those scores. In my opinion the conspicuity scores should be done by at least four observers to have more robustness and more conclusive results on that part.
  2. Related to that point, the Diagnostic T2w MRI was not optimized to achieve an improved contrast for the urethra. This should be properly discussed.
  3. Images on the built pdf document are small. Is it not possible to provide high resolution and large images to perform a proper evaluation of the image quality?
  4. Figure 4 is difficult to interpret because of the presence of delineations from other image datasets not included in the figure and the registration of the CT to the HASTE. In addition, it is not reported in the manuscript how the registration was done. On the prostate? Bone registration? On the urethra? Depending on the registration performed a different result is obtained regarding to the agreement of the urethra delineations.
  5. Please, indicate which quartile range was used in the boxplots. Does the red line represent the Median value?
  6. On page 10, line 287 it is said that the differences in contours might be due to potential motion during long acquisition. Did the authors also acquired a 3D bSSFP scan before and after? Such a scan can be acquired quite fast and might help to ascertain patient motion during the session.
  7. Page 10, line 300: could you provide the PE direction of the different protocols? It could be included in the table about the sequence protocols.
  8. It is reported in the manuscript that previous studies have expanded the urethra contour isotropically by 2 mm. This is probably always needed as uncertainty in the delineation and intra-fraction motion management margin. Can the authors provide some insight about their uncertainty in the delineation? Or at least, discuss which margin they consider necessary for radiotherapy purposes?
  9. In MRgRT usually a bSSFP protocol is used for "real-time" MRI. is it feasible to fine-tune a bSSFP protocol for "real-time" MRI in order to visualize the urethra during irradiation? This could be included in the discussion.
  10. Small textual suggestions:
    1. Page 3, Table 1: please, remove "m" from Echo Time row at the last column.
    2. Page 3, Table 1: please, include degree symbol in Flip Angle row at the last column.
    3. Page 3, line 100, Page 4, line 172: isotopically -> isotropically

Reviewer 2 Report

Paper of interest for institutions treating prostate cancer patients with MR-LINAC. This paper demonstrates the difficulties visualizing organs at risk, eg. urethra, and using the full potential of MRgRT to spare critical structures.

Suggestions for improvement:

1. Introduciton

It would be helpful for the reader to explain what these new proposed MR sequences (3D HASTE and 3D TSE sequences) provide as additional information based on what physical/physiological background compared to standard MR sequences. What did you opitmize? What are the disadvantages of these sequences?

3. Results

I suggest you move 3.1 ot Material and Methods or you mention in Material and Methods that healty volunteers were part of this study

Table2: I propose to add a row showing the differece between the different imaging protocols and the p-values

4. Discussion

The finding that optimized MRgRT 3D HASTE and TSE showed less motion/ghosting artifacts, originating from subcutaneous fat and periprostatic fat, and better urethra/prostate contrast than clinical MRgRT 3D bSSFP is interesting. However, according to table 1 it adds significant time for the patient lying on the couch. You should comment how prolonged treatment time is justifyed and how this disadvantage for the patient may turn into better treatment outcome. Unfortunately, you did not report any GU toxicity for 11 study patients. Idealy, this study should have been done with a cohort using MRgRT 3D bSSFP only and then compare them with the use of additional MRgRT sequences. Lack fo this data limits the impact of the presented study. You may comment on this. 

Prolonging treatment time, does this have negativ effects on the patients positioning, bladder filling, prostate motion etc.? You may comment in the discussion.

What about prostate cancer tumors close to the urethra or in the transitional zone, would you recomment urethral sparing in all cases? Please make a comment how you decide when to spare and how reliable current MRgRT sequences are to address this question.

If you detect prostate motion during treatment (intrafraction motion) which of the presented MRgRT sequence would you use to correct, please comment.

Reviewer 3 Report

SBRT for prostate cancer is an important innovative technique and urethra definition an important subject for the radiation oncologist. I see the manuscript to be appropriate for a specialized radiation oncology journal.

Specific problems: A small number of patients were considered with only a single senior observer. Without a gold standard, a comparison of different sequences is difficult. It is well known that urethra cannot be defined in CT. I am not well convinced that the urethra position within the prostate varies considerably between fractions.

Round 2

Reviewer 2 Report

No further comments. The authors addressed my suggestions and adapted the manuscript accordingly

Reviewer 3 Report

The manuscript has been improved in the methods, results and discussion, in particular by additional observers.